# Learning Riemannian metric for disease progression modeling

**Samuel Gruffaz**
Inria Paris
Ecole normale supérieure Paris-Saclay
`samuel.gruffaz@ens-paris-saclay.fr`

**Pierre-Emmanuel Poulet**
Paris Brain Institute
Inria Paris
`pierre-emmanuel.poulet@inria.fr`

**Etienne Maheux**
Paris Brain Institute
Inria Paris
`etienne.maheux@icm-institute.org`

**Bruno Jedynak**[*]
Maseeh Professor of Mathematical Sciences
Fariborz Maseeh Hall, room 464P
Portland, OR 97201
`bruno.jedynak@pdx.edu`

**Stanley Durrleman**[†]
Paris Brain Institute
Inria Paris
Inserm, CNRS, Sorbonne University
`stanley.durrleman@inria.fr`

for the Alzheimer's Disease
Neuroimaging Initiative [‡]

## Abstract

Linear mixed-effect models provide a natural baseline for estimating disease progression using longitudinal data. They provide interpretable models at the cost of modeling assumptions on the progression profiles and their variability across subjects. A significant improvement is to embed the data in a Riemannian manifold and learn patient-specific trajectories distributed around a central geodesic. A few interpretable parameters characterize subject trajectories at the cost of a prior choice of the metric, which determines the shape of the trajectories. We extend this approach by learning the metric from the data allowing more flexibility while keeping the interpretability. Specifically, we learn the metric as the push-forward of the Euclidean metric by a diffeomorphism. This diffeomorphism is estimated iteratively as the composition of radial basis functions belonging to a reproducible kernel Hilbert space. The metric update allows us to improve the forecasting of imaging and clinical biomarkers in the Alzheimer's Disease Neuroimaging Initiative (ADNI) cohort. Our results compare favorably to the 56 methods benchmarked in the TADPOLE challenge.

---

[*]This work was partially supported by NIH R01AG027161 and R01EY032284.

[†]This research has also received funding from the program "Investissements d'avenir" ANR-10-IAIHU-06. This work was also funded in part by the French government under management of Agence Nationale de la Recherche as part of the "Investissements d'avenir" program, reference ANR-19-P3IA-0001 (PRAIRIE 3IA Institute) and by ANR under the joint programme in neurodegenerative diseases (JPND) ANR-19-JPW2-000 (E-DADS).

[‡]Data used in preparation of this article were obtained from the Alzheimer's Disease Neuroimaging Initiative (ADNI) database (adni.loni.usc.edu). As such, the investigators within the ADNI contributed to the design and implementation of ADNI and/or provided data but did not participate in analysis or writing of this report. A complete listing of ADNI investigators can be found at: http://adni.loni.usc.edu/wp-content/uploads/how_to_apply/ADNI_Acknowledgement_List.pdf

# 1  Introduction

Understanding the progression of diseases is essential to accurately monitor, diagnose and predict patients' state of health. Disease progression modeling analyses longitudinal data to capture common and subject-specific progression patterns. Longitudinal data analysis has usually been addressed in the framework of parametric mixed-effects models [1]. The population parameters express the characteristics of the disease, and the individual parameters encode the specificities of each patient. It represents the diversity of pathology. The challenge is to construct models flexible enough to learn the disease heterogeneity across the population and sufficiently interpretable to derive a practical conclusion.

To understand the trade-off between flexibility and interpretability, we review these models through the prism of time and space variability of patients' trajectories. We refer to time variability when all the patients follow the same trajectory but not at the same speed and not with the same onset and to space variability when each patient follows a different trajectory but with the same speed and the same onset. For instance, to observe the different stages of a disease, event-based models have been introduced [2], the timeline of disease progression is seen as a succession of stages followed by each patient but not in same order. Time variability is encoded into this subject-specific ordering. Extension to mixture model includes space variability as a set of possible trajectories[3]. Increasing the number of stages and mixture components improves both the space and time granularity but it will result in a lack of interpretability of each stage.

To express more finely the space granularity, we can assume the patients' observations follow a continuous parametric curve, the dataset providing discrete, noisy observations along this curve. [4, 5, 6]. These curves can be defined with mechanistic approaches using Ordinary Differential Equations (ODE), the sources of variability (time and space) across patient being in the Cauchy conditions ($t_0$ and $X(t_0)$) and the common pattern of progression in the vector field as derivative [4, 5]. This family of models is interesting to study the correlation between time and space variability [4], but it is a double-edge sword since these two factors cannot be analysed independently. Probabilistic approaches to represent parametric curves can solve this point. Gaussian Processes (GP) have been applied for disease progression successfully [7] to study correlation in the pathological progression while keeping time and space variability separated. Their semi-parametric paradigm allows for a significant flexibility by encoding the space-variability in the Gaussian kernel, but this last part is not easy to interpret. More generally in Bayesian models, it is difficult to disentangle the different random effects to obtain meaningful parameters. To address this point, geometric approaches have proved to be efficient for scalar biomarkers [6] and brain shapes [8].

One instance of this family of geometric models [6] assumes that each subject follows a curve on a Riemannian manifold which is a translation from a common geodesic (Disease Course Mapping, DCM). The space variability is encoded in the translation and the time variability by a time reparametrization. Though the Riemannian formalism provides interesting tools for modeling trajectories, it is usually constrained to well-studied manifolds with simple geodesics such as sigmoid curves for biomarkers or straight lines for cortical atrophy. In order to increase the model flexibility, some contributors propose to learn the metric by playing on the representation of trajectories [9]. They rely on a push-forward method to learn a metric with pseudo-diffeomorphic transformations [10]. Observations' trajectories are seen as transformations of straight lines evolving in a latent space learned by an auto-encoder. Though this semi-parametric representation enables multiple progression profiles, it is inappropriate for disentangling space and time variability.

Another method for trajectory learning is to use the theory of shape analysis and more especially the deformation of shapes in time. Authors commonly learn the deformation as a diffeomorphism with a Reproducing Kernel Hilbert Space (RKHS), the kernel regularity encoding the smoothness of the deformation through time [8, 11]. We believe that this deformation method is an interesting alternative to the the auto-encoders to increase model flexibility while keeping control on space and time variability.

In this work, following [9] but taking a step back, we propose a semi-parametric method using a RKHS to learn the Riemannian metric of DCM models. We thereby retain the possibility to learn the inter-variability of patient trajectories thanks to the mixed-effect framework, keeping the advantages of the geometric and Bayesian approach presented in [12]. First, we recall the structure of the mixed-effect model in section 2.1, then we present the method to learn the metrics step by

step in section 2.2. We validate the presented method on synthetic data in 3.1 and on a real dataset (ADNI,Tadpole) in 3.2 by comparing it with previous models on the task of predicting patient's biomarker progression. Finally, we discuss limitations and possible future works in the discussion section 4.

## 1.1 Related works

Recurrent neural networks have been proposed to predict a sequence of biomarker values. Such methods, however, need to cope with missing values, a limited and variable number of observations per patient, a variable time interval between observations, and a very variable pattern of change depending on disease stage and subject.

Mixed-effect models make several assumptions about the pattern of progression. In particular, they assume that the progression falls within a parametric family of curves of small dimension, e.g. a linear curve or a logistic curve with unknown parameters. This is the case for instance of the DCM approach where the linear or logistic curve are seen as the solution of an Hamiltonian flow (a geodesic is a particular case of an Hamiltonian flow) with a given Hamiltonian function. Our main contribution is to learn the metric of the Riemannian manifold, and therefore the Hamiltonian, to better learn the dynamics of progression of the biomarker. In this respect, our contribution relates to the techniques of Hamiltonian and normalizing flows that are becoming popular in the machine learning community, see e.g. [13] or [14].

## 2 Methodology

All proofs are given in the Supplementary material A. Notations in all the following are: $|.|$ for the usual Euclidean norm, $\langle ., . \rangle$ the Euclidean scalar product, $|||L||| = \sup_{x \in (\mathbb{R}^d)^*} \frac{|L(x)|}{|x|}$ for all linear function $L$ and $||| \, \mathrm{d} \, f |||_\infty = \sup_x ||| \, \mathrm{d} \, f(x) |||$ for all differentiable function $f$.

### 2.1 Riemannian geometric setting of the mixed-effect model.

In this part, we present the geometrical construction of the mixed-effect models [6] with further explanations in Supp. material A. Let $\mathbb{M}$ be an open subset of $\mathbb{R}^d$ ($d$ is the dimension of the observations) equipped with the Riemannian metric $g$. We make now explicit the dependency on the metric $g$ that will be optimized.

**The mixed-effect model.**    Based on [12], observations are seen as a point in a Riemannian manifold $\mathbb{M}$, denoted $y_{i,j}$, corresponding to the data of the $i$-th subject at its $j$-th visit at time $t_{i,j}$. The points are seen as noisy samples along an individual trajectory $\gamma_i^g$, namely a curve on the manifold, which result of a random spatiotemporal transformation of a reference geodesic $\gamma_0^g$ on the manifold. The general formula states:

$$y_{i,j} = \underbrace{\eta_{w_i}^g(\gamma_0^g)(\psi_i(t_{i,j}))}_{=\gamma_i^g(t_{i,j})} + \epsilon_{i,j}, \;\; \epsilon_{i,j} \sim \mathcal{N}(0, \sigma^2 I_d) \text{ i.i.d}, \; \sigma > 0 \tag{1}$$

- $\psi_i : t \rightarrow \alpha_i(t - t_0 - \tau_i) + t_0$ is a time-reparametrizing function, with $t_0$ being a reference time for the population. The individual variability in time is captured by the two subject-specific parameters in the time-shift $\tau_i$ and the acceleration factor $\alpha_i$. The time-shift $\tau_i$ represents the delay at onset relative to $t_0$ for the individual i, accounting for early or late onset. The $\alpha_i$ models the speed at which the trajectory of individual i is traveled, thus accounting for fast or slow progressors of the disease.
- $\gamma_0^g : t \rightarrow \mathrm{Exp}_{p_0,t_0,t}^g(v_0)$, the population average trajectory, is the geodesic passing at point $p_0$ with velocity $v_0$ at time $t_0$. It represents the fixed effects of the mixed-effects model. This supposedly summarizes the characteristic progression of the disease at hand.
- $\eta_{w_i}^g(\gamma_0) : s \rightarrow \mathrm{Exp}_{\gamma_0(s),s,1}^g(P_{(\gamma_0,t_0,s)}^g(w_i))$ is the exp-parallelisation of the geodesic $\gamma_0$ in the subject-specific direction $w_i$ called space-shift. We denote $P_{(\gamma_0,t_0,s)}^g(w_i)$ the parallel transport of the vector $w_i$ along the curve $\gamma_0$ from $\gamma_0(t_0)$ to $\gamma_0(s)$. The notion of exp-parallelisation generates the spatial inter-variability among the population (disentangled

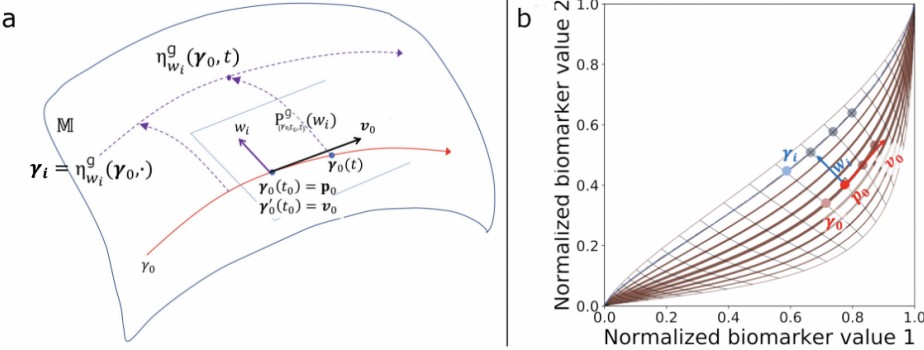

Figure 1: a) sketch of exp-paralellisation in an abstract manifold. b) exp-parallelisation for the unit square as a manifold with the metric $g_{(p_1,p_2)}((x_1,x_2),(y_1,y_2)) := \frac{x_1 y_1}{p_1^2(1-p_1)^2} + \frac{x_2 y_2}{p_2^2(1-p_2)^2}$. The average trajectory in red and an individual trajectory in blue or purple.

from the time-variability encoded in $\psi_i$), it is depicted in (1) and further developed in the Supp. material A. The space-shift $w_i$ encodes the specificity of the individual biomarkers' evolution, to help distinguish between different patterns of disease progression.

In the mixed-effects model framework, we identify the random effects as the individual parameters $z_i = (w_i, \tau_i, \alpha_i)$ for the patient $i$. They induce a spatiotemporal transformation of the average trajectory parametrized by the fixed effects $\theta_0 = (p_0, v_0, t_0)$. To ensure a unique decomposition of the temporal and spatial effects, the $w_i$ are chosen orthogonal to $v_0$ in the tangent space at $p_0$. Normal distributions are chosen as priors for $\tau_i, w_i$ and $\xi_i$ with $\alpha_i = \exp(\xi_i)$. All the model effects can be estimated by maximizing the likelihood with the algorithm MCMC-SAEM [15, 16] providing the convergence guarantees (see [12] for the associated statistical specification).

**The metric.** Traditionally, the metric $g$ is selected using empirical arguments. In contrast, we propose to estimate $g$ from data, thus relieving from a potentially tricky modeling task. This last parameter is the most important since the whole model depends on it: it encodes the shape of geodesics with $\mathrm{Exp}^g$ and the variability due to exp-parallelisation with $\eta^g$. In [12], [6], the authors choose $(\mathbb{M}, g)$ as a manifold product of dimension-1 manifold to have a closed form for $\mathrm{Exp}^g$ and $\eta^g$. For example, we can choose $\mathbb{M} = \mathbb{R}^d$ and $g = \langle .,. \rangle$ to recover straight lines as geodesics or $\mathbb{M} = (0,1)^d$ and $g_p = \langle f_p(.), f_p(.) \rangle$ with $f_p(v) = (\frac{v_i}{p_i(1-p_i)})_i$ to recover logistic curves as geodesics. Product metrics assume that the geodesics coordinates are independent which is not necessarily true when they describe neurodegenerative bio-markers progression [17]. Moreover, due to the product-metric, the coordinates follow geodesics which enforces bijectivity and thus monotonicity of the trajectories. Thus, we assume that the noise effect $\epsilon_{i,j}$ in (1) is in reality different from the real data noise $\epsilon_{i,j}^{\mathrm{noise}}$ because of the miss-specification of the metric $g$, in such a way that $\epsilon_{i,j} = \epsilon_{i,j}^g + \epsilon_{i,j}^{\mathrm{noise}}$ with $\epsilon_{i,j}^g$ the error due to the modelling. The goal is to minimize $\epsilon_{i,j}^g$.

To state the problem, we suppose that the population and individual parameters $\theta = (\theta_0, (z_i)_i)$ are estimated and we want to minimize the squared reconstruction error $L(g)$ according to $g$:

$$L(g) := \sum_{i,j} |\epsilon_{i,j}|^2 = \sum_{i,j} |\epsilon_{i,j}^g + \epsilon_{i,j}^{\mathrm{noise}}| = \sum_{i,j} |y_{i,j} - \gamma_i^g(t_{i,j})|^2 \qquad (2)$$

To prevent the risk of overfitting since $\epsilon_{i,j}^{\mathrm{noise}}$ is present in L, we can add a regularizing term or constrain the set of admissible metrics $g$. Next, we describe a method to solve this optimization problem.

## 2.2 Learning the metric.

The task of learning a Riemannian metric has already been addressed in the literature [10],[9], [18] and most of the time, it is reduced to the task of learning a diffeomorphism using the following proposition (method coming from [10]):

**Proposition 1** (Pushforward metrics). *Provided $(\mathbb{M}, g)$ a Riemannian space, $N$ a manifold and $\phi : \mathbb{M} \to N$ a $C^1$ diffeomorphism, we can equip $N$ with the Riemannian metric $g^\phi$ defined as:*

$$\forall\, p \in N, \ \ \forall w, v \in TN_p, \ \ g_p^\phi(w, v) = g_{\phi^{-1}(p)}(\mathrm{d}\,\phi^{-1}(p).w, \mathrm{d}\,\phi^{-1}(p).v)$$

*Moreover, $\phi$ is an isometry, which implies that for all $(m, v)$ in $T\mathbb{M}$ and $\gamma : (-1, 1) \to \mathbb{M}$ a differentiable curve:*

$$\mathrm{Exp}_{\phi(m)}^{g^\phi}(\mathrm{d}\,\phi(m).v) = \phi \circ \mathrm{Exp}_m^g(v), \ P_{\phi \circ \gamma, t_0, s}^{g^\phi}(\mathrm{d}\,\phi(\gamma(t_0)).v) = \mathrm{d}\,\phi(\gamma(s)).P_{\gamma, t_0, s}^g(v)$$

Thanks to this property, we can start from an initial metric $g$ in $\mathbb{M}$ and reach a wide variety of Riemannian metrics by considering $(\phi(\mathbb{M}), g_0^\phi)$ with $\phi$ in $\mathcal{F}$ a set of $C^1$-diffeomorphism. Moreover, the exp-parallelisation $\eta^{g^\phi}$ can be expressed in closed form as soon as $\phi$ and $\eta^g$ are expressed in closed form as well. Nevertheless, it restricts the possibilities because the curvature is preserved. For example if $g$ is the Euclidean metric and $\mathbb{M} = \mathbb{R}^d$, the space $(\phi(\mathbb{M}), g^\phi)$ will be always flat (curvature equals zero) since $(\mathbb{M}, g)$ is flat.

Learning a diffeomorphism is a more common task than learning a metric especially in the field of shape analysis with the LDDMM algorithm [11],[8] and even in deep learning with the invertible networks [19] for different applications [9],[20],[21]. Inspired from the LDDMM framework, we chose to use the following method to construct the set of $C^1$-diffeomorphism $\mathcal{F}$:

**Proposition 2.** *If $\phi = \mathrm{id} + f$, with $f$ a bounded function in $C^1(\mathbb{R}^d, \mathbb{R}^d)$ such that $|||\,\mathrm{d}\,f|||_\infty < 1$ (H1), then $\phi$ is a $C^1$ diffeomorphism.*

In all the following we choose $\mathbb{M} = \mathbb{R}^d$ which is the most flexible choice. For example if we want to subsequently work on $]0, 1[$, consider $\phi' = \phi \circ \sigma$ with $\sigma = \frac{1}{1 + \exp(-t)}$. To respect the condition of the previous proposition, we choose to take $f$ in an RKHS denoted $\mathcal{H}$ to encode its regularity in its norm. Moreover, the RKHS functions are handy in non-parametric optimization thank to the representer theorem (for a comprehensive review of kernel methods [22]). We derived the following lemma to fit the constraints:

**Lemma 1.** *If $k$ is a kernel such that $k(x, y) = g(x-y)\,\mathrm{Id}$ with $g \in C^2(\mathbb{R}^d, \mathbb{R})$ and $h(x, y) = g(x-y)$ is a bounded kernel, then $\forall f \in \mathcal{H}$ s.t $||f||_\mathcal{H} < \frac{1}{\sqrt{|\nabla g(0)|}}$, $|||\,\mathrm{d}\,f|||_\infty < 1$ and $f$ is bounded (H1).*

*Examples:*

- *The Gaussien kernel; $g(x) := \exp(-\frac{|x|^2}{2\sigma^2})$, $\sigma > 0$ $\frac{1}{|\nabla g(0)|} = \frac{\sigma}{\sqrt{d}}$.*

- *The Sobolev kernel or generalized T-student kernel, $g(x) = \frac{1}{(1 + \frac{|x|}{2\sigma^2})^a}$, $\sigma > 0, a > d$, $\frac{1}{|\nabla g(0)|} = \frac{\sigma}{a\sqrt{d}}$*

In all the following we choose a kernel $k$ respecting the previous conditions such that for all $f$ in the closed ball $\bar{B}_\mathcal{H}(0, c)$ (c a positive constant), (H1) is verified. Note that we can prevent overfitting thanks to this constraint. Now, we are equipped to address the optimization problem (2).

**Lemma 2.** *If the population and random effects $\theta = (\theta_0, (z_i))$ are estimated, the minimization of the error of reconstruction (2) can be performed by solving the following problem:*

$$f^* \in \mathrm{argmin}_{f \in \bar{B}_\mathcal{H}(0,c)} L(f), L(f) = \sum_{i,j} |y_{i,j} - \gamma_i(t_{i,j}) - f(\gamma_i(t_{i,j}))|^2 \tag{3}$$

*Thanks to the representer theorem, we have:*

$$\forall x \in \mathbb{M}, f^*(x) := \sum_{i,j} k(x_{i,j}, x) w_{i,j}, x_{i,j} = \gamma_i(t_{i,j})$$

*With this parametric form for $f^*$, the previous problem (3) is equivalent to:*

$$\underset{W}{minimize} \qquad ||U - K_X W||^2 \tag{4a}$$

$$subject\ to \qquad W^T K_X W \le c_0, \tag{4b}$$

*where the previous variables are vectorized:*

$$W := (w_{i,j})_{i,j}, \ U := (y_{i,j} - \gamma_i(t_{i,j}))_{i,j}, \ K_X := (k(x_{i,j}, x_{k,l}))_{((i,j),(k,l))}$$

---

**Algorithm 1** Geodesics Bending (Alternating maximization algorithm)

---

**Require:** $g_{\text{init}}, \theta_{\text{init}}, N_{\text{comp}}, k, N_c, n_{\text{MCMC}}$

    $g \leftarrow g_{\text{init}}; \ \phi \leftarrow \text{id}; \ \theta \leftarrow \theta_{\text{init}}$

    **for** $l = 1$ to $N_{\text{comp}}$ **do**

        Run the MCMC-SAEM for $n_{\text{MCMC}}$ iterations with metric $g$ to estimate $\theta_l$

        $\theta \leftarrow \theta_l$.

        Solve the optimization problem (4a) with parameters $\theta$ to estimate $f_l^*$

        $\phi \leftarrow (\text{id} + f_l^*) \circ \phi$

        $g \leftarrow g^\phi$

    **end for**

    **return** $\theta, g$

---

The minimization of a quadratic function under constraints (4a) can be solved numerically within a reasonable time as soon as the matrix $K_X$ is not in high dimensions. The total number of visits in a medical cohort can reach up to dozens of thousands and the points $x_{i,j}$ are close to each other, which results in the matrix being ill-conditioned. To overcome these obstacles, we assume that $y_{i,j} \in [0.1]^d$ and we create a grid of control points in $[0,1]^d$ with step-size $\sigma$ if $\sigma$ is the kernel variance (there exists other methods [23], but this one is more close to the LDDMM framework [11] and gives satisfying results [24]). We only keep the grid points which are $\sigma$-close to the observations, thus removing the useless ones. We note $N_c$ the number of control points, which can increase exponentially with the dimension according to the previous construction: $N_c(\sigma) \approx \frac{1}{\sigma^d}$. If $d$ is too large, we can choose another method to obtain a reasonable number of control points, a simple way to do so is to subsample the points $(x_{i,j})$ [23]. In all the following, the grid method is used because of its practical upsides (no randomness, no risk of ill-conditioning of the matrix $K_X$).

Now, we have a method to improve the choice of the metric when keeping the mixed-effect parameters $\theta$ fixed. Then we need to both optimize the metric $g$ and the model parameters $\theta = (\theta_0, (z_i))$. We choose to maximize the likelihood with algorithm 1. As we know how to optimize $g$ when $\theta$ is fixed and reciprocally, we implement an alternating scheme of optimization. We start with a simple metric $g_{\text{init}}$ (usually the Euclidean metric or the metric resulting from the push-forward of the logit), and we estimate the mixed-effect parameters $\theta$ with the MCMC-SAEM algorithm [15, 16]. Then, with $\theta$ fixed, we estimate the metric by numerically solving the optimization problem (4a) with the chosen kernel $k$ and a chosen number of control points $N_c$. Each iteration is a repetition of these two steps. Each step with a metric optimization amounts to a composition of the previously estimated diffeomorphism $\phi$ with the newly estimated diffeomorphism. The number of iterations thus amounts to the number of compositions $N_{\text{comp}}$. The hyperparameters $k$, $N_{\text{comp}}$ and $N_c$ are tuned to make a compromise between loss minimization, computation time and number of parameters.

For $N_{\text{comp}}$ compositions, we have $g = g^{\Phi_{N_{\text{comp}}}}$ with $\Phi_{N_{\text{comp}}} = \phi_{N_{\text{comp}}} \circ \ldots \circ \phi_1$ where $\phi_i = \text{id} + \sum_{j=1}^{N_c} k(x_j^i, .) w_j^i$, $(x_j^i)_j$ the $N_c$ control points and $(w_j^i)_j$ their associated weights. This structure resembles the deep neural networks considering $N_c$ as the width and $N_{\text{comp}}$ as the depth. The total time complexity is $\mathcal{O}(n_{\text{MCMC}} d N_c N_{\text{comp}}^3)$ which reduces in practice the choice of $N_{\text{comp}}$ (see Supp. material B to more details on these considerations). We called this method Geodesics Bending (GB).

## 3 Experiments

All the methods are developed in Python by extending the open-source Leaspy library (https://leaspy.readthedocs.io) created for DCM models and run on a 2.80GHz CPU with 16 GB RAM. All code will be available on Github in the near future.

### 3.1 Synthetic data

In this part, we study the learning capacity of the method and its stability on synthetic data depending on the values of the depth $N_{\text{comp}}$, the width $N_c(\sigma)$ and the initial metric $g_{\text{init}}$. We study the algorithm in the context of neurodegenerative diseases. The parameters are selected to be realistic.

In all the following, to perform the metric estimation, we choose $k$ to be the Gaussian kernel (experiments with the Sobolev kernel gives similar results), $n_{\text{MCMC}} = 200$ ($n_{\text{MCMC}} = 10000$ at the

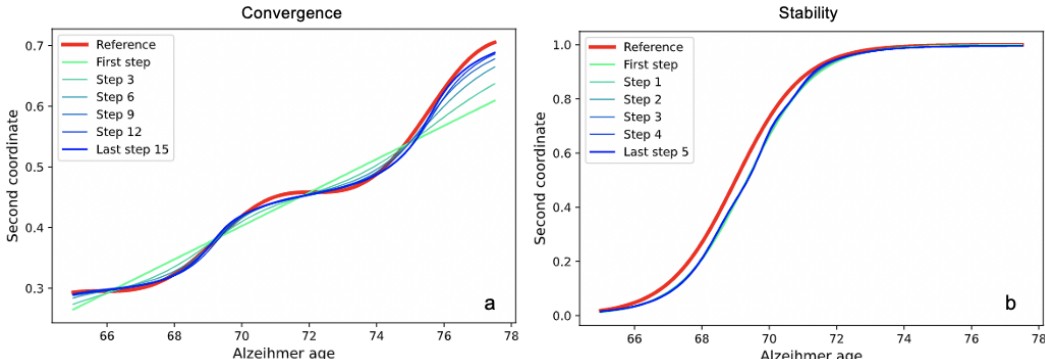

Figure 2: (a) The average trajectory on the second coordinate is displayed for each value of $N_{\text{comp}}$ in the Convergence experiment. (b) The average trajectory on the second coordinate is displayed for each value of $N_{\text{comp}}$ in the Stability experiment.

very first step since the mixed-effects are not necessarily well initialized whereas for subsequent iterations the MCMC-SAEM starts closer to the optimum), $\theta_{\text{init}}$ is computed empirically from patients features [25]. It leaves $N_{\text{comp}}, N_c(\sigma)$ and $g_{\text{init}}$ to select according to the situation.

**Data generation**   Once we have selected the number of patients $N_{\text{pat}}$ and the observations' dimension $d$, we generate data according to the DCM model (1) selecting the fixed effects $t_0, v_0, p_0, g_{\text{gen}}, \sigma_\tau^2, \sigma_\xi^2, \sigma_{\text{noise}}^2$, and sampling the random effects parameters such that $\tau_i \sim \mathcal{N}(0, \sigma_\tau^2), \xi_i \sim \mathcal{N}(0, \sigma_\xi^2), \epsilon_{i,j} \sim \mathcal{N}(0, \sigma_{\text{noise}}^2)$, principal directions of variation for space-shifts $(A_m)$, $s_i^m \sim \mathcal{N}(0,1), w_i = \sum_m s_i^m A_m$ and time parameters $\sigma_t, \mathrm{n}_{\min}^t, \mathrm{n}_\delta^t$ to generate the patient time-points $(t_{i,j})_{1 \leq j \leq T_i}$ randomly: $T_i = \mathrm{n}_{\min}^t + \delta_i, \delta_i \sim B(n_\delta, \frac{1}{2}), t_{i,j} \sim \mathcal{N}(t_0, \sigma_t), y_{i,j} = \gamma_i^{g^\phi}(t_{i,j}) + \epsilon_{i,j}$. For both experiments, we set $N_{\text{pat}} = 500, d = 2, \mathrm{n}_{\min}^t = 2, \mathrm{n}_\delta^t = 6, \sigma_t = 4$.

**Convergence**   First, we experiment whether learning the metrics with kernels enables the regression of exotic trajectories. We choose $\sigma_{\text{noise}} = 0.01, g_{\text{gen}} = g^\phi$ with $\phi(x) := \exp(0.07(x + \sin(x)) - 1)$ and $g = \langle ., . \rangle$ for the data generation and $N_c(\sigma = 0.08) \approx 90$ for the metric estimation (more details in Supp. material C).

Increasing the number of compositions improves the capacity to recover the true average trajectory as we can see in figure 2. The estimated noise variance $\tilde{\sigma}_{\text{noise}}^2$ decreases linearly and plateau around the true value of noise $\sigma_{\text{noise}}^2 = 0.013$ which shows that the model is a little under-fitting. The kernel regularization impacts the smoothness of the function, making it harder to fit the fast oscillations. By testing different values of the width $N_c(\sigma)$, we observed that the lower the kernel variance $\sigma$, the higher the risk to overfit. Conversely, the higher $\sigma$, the higher the risk to underfit. As for deep neural networks, there is an optimum to find regarding the hyper-parameters. In practise with datasets on neurodegenerative disease, normalized data are very noisy $\sigma_{\text{noise}}^2 \approx 0.05$. To avoid over-fitting, we select the kernel variance $\sigma$ in [0.1,0.5]. The choice of $g_{init}$ influences the number of compositions required to reach a good approximation of the "true" metric.

**Stability**   Secondly, we experiment whether the method is stable: beginning from the metric which has generated the data with $g_{init}$, we observe the effect of supernumerary compositions. We choose $g_{\text{gen}} = g^{\text{sig}}$ with $\text{sig}(x) := \frac{1}{1+\exp(-x)}$ and $\sigma_{\text{noise}} = 0.05$ for the data generation, $g_{init} = g^{\text{sig}}$ and $N_c(\sigma = 0.2) \approx 31$ for the metric estimation.

The compositions on $\phi$ produce local fluctuation on the sigmoid trend as pictured in figure 2. There are less observations near the inflexion point which causes over-fitting. This phenomenon is also present in datasets on neurodegenerative disease: areas of the trajectory where data are scarce can be over-fitted by GB. Nevertheless, the value of the effects $\theta$ are nearly constant after its first estimation in the algorithm 1 (see Supp. material C), which strengthens the model interpretability and encourages the reduction of $n_{\text{MCMC}}$.

Table 1: Experiment performances recorded with the MAE. The proposed method is compared with the alternative using a paired, two-sided Wilcoxon signed rank test. *: p<0.05, **: p<0.01,***: p<0.001.

| | MMSE | | HIPP | | VENTS | | ABETA | |
| --- | --- | --- | --- | --- | --- | --- | --- | --- |
| Experiment | GB | DCM | GB | DCM | GB | DCM | GB | DCM |
| Pred($n_f = 1$): | 0.140 | 0.146 | **0.046**\*\*\* | 0.056 | **0.042**\*\*\* | 0.056 | 0.124 | 0.114 |
| Pred($n_f = 2$): | **0.132**\*\* | 0.146 | **0.037**\*\*\* | 0.046 | **0.038**\*\*\* | 0.049 | 0.123 | **0.109**\* |
| Imp($n_r = 1$): | **0.080**\*\* | 0.088 | **0.029**\*\*\* | 0.31 | **0.017**\*\*\* | 0.020 | 0.128 | 0.123 |
| Imp($n_r = 2$): | **0.081**\*\*\* | 0.089 | **0.028**\*\*\* | 0.030 | **0.023**\*\*\* | 0.027 | 0.120 | **0.111**\*\* |

**The problem of generalization**   The method reveals to be flexible and quite stable provided the hyper-parameters are wisely selected. To assess whether the learning complexity of the method makes sense in practice, we propose to measure its capacity of generalization by recording its performance on prediction tasks within a 5-folds cross validation framework: running the algorithm 1 on 4-folds, predicting on the last fold the last patient visit from its first visits, or predicting the previous visits from the last visits (as for data imputation). In the next part, these experiences are carried out on a reference dataset of neurodegenerative diseases with different types of data (cognitive scores, sub-cortical volumes from MRI, cerebro-spinal fluid biomarkers) to show the potential of a flexible method such as the one we introduce in this paper.

## 3.2   Real data

**Introduction ADNI**   The data used in this article were obtained from the Alzheimer's Disease Neuroimaging Initiative (ADNI) database (adni.loni.usc.edu), launched in 2003 as a public-private partnership, led by Principal Investigator Michael W. Weiner, MD. The primary goal of ADNI has been to test whether serial magnetic resonance imaging (MRI), positron emission tomography (PET), other biological markers, and clinical and neuropsychological assessment can be combined to measure the progression of mild cognitive impairment (MCI) and early Alzheimer's disease (AD).

First, the algorithm was tested on the subset of AD patients. For each visit, we jointly model their Mini-Mental State Examination (MMSE, a cognitive score), the volume of their hippocampi and lateral ventricles normalized by their intracranial volume (HIPP and VENTS respectively) and the concentration of amyloïd $\beta_{1-42}$ in their cerebro-spinal fluid (ABETA). The goal was to understand the impact of the metric estimation on these typical AD biomarkers. Then, the model was compared to the state-of-the-art with The Alzheimer's Disease Prediction Of Longitudinal Evolution (TADPOLE) challenge [26]. These methods include penalized regression, linear mixed-effect models, recurrent neural networks, and multi-task learning. Since the challenge was closed, more algorithms have been presented for the same prediction task, using the same dataset, including [27] and [28] using deep RNNs, and [29] using random forests. The forecast of ventricle volumes (VENTS) and cognitive decline (ADAS-cog) presented in this paper have smaller mean absolute errors than all these competing methods.

**An Alzheimer's disease cohort**   The cohort is composed of patients from all ADNI phases (ADNI1, ADNI GO, ADNI2, ADNI3), having been diagnosed with AD, with a minimum of 2 visits (4.15 visits on average and 1.5 visits for standard deviation) and ABETA < 977 pg/mL cutoff (so to get amyloid-positive patients only [30, 31]). Sub-cortical volumetric segmentations of MRI were performed with the Freesurfer image analysis suite v6.0.0, which is documented [32] and freely available for download online (http://surfer.nmr.mgh.harvard.edu/). All scores were normalized in [0,1] (to motivate the logistic prior for trajectories), using preprocessing tools from the scikit-learn library [33]: MMSE was affinely-mapped using the score bounds, other biomarkers were Box-Cox transformed [34] to un-skew their distribution, clipped between their 0.1 and 99.9 percentiles and finally affinely-mapped into [0,1]. When needed, scores were reversed such that they all increase during disease progression. The models being generative, we did not drop out missing values. We compare the DCM models using the Riemannian metric $g^{\text{sig}}$ (baseline) and GB ($g_{\text{init}} = g^{\text{sig}}$, $N_{\text{comp}} = 6$ and $N_c(\sigma = 0.24)$), both using 2 principal directions of variation for space-shifts. Note that the baseline (DCM in yellow in figure 3) is the very first iteration of GB optimization.

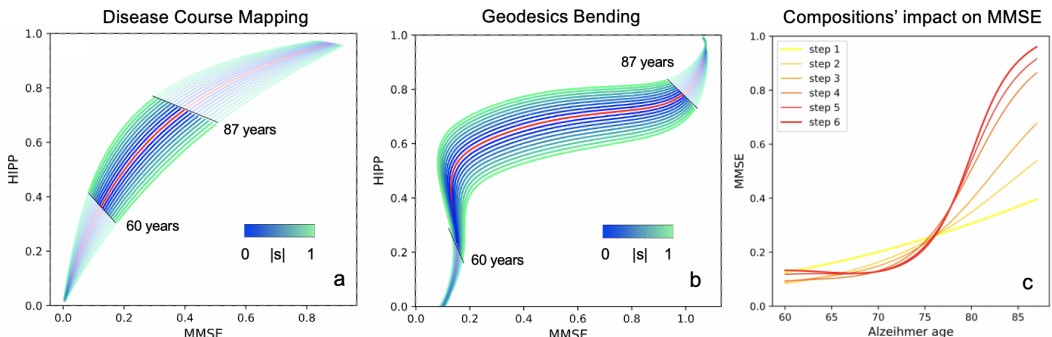

Figure 3: (a, b) The estimated tubular coordinate system is shown for the HIPP and MMSE scores, showing the distribution of the spatial variability around the average geodesics in red by changing the space-shifts on a principal direction ($w_i = sA_2, s \in [-1, 1]$). (c) The effects of each learning step on the average trajectory $\gamma_0$ is shown. 60 and 87 years are the 0.05 and 0.95 quantile for the ages ($t_{i,j}$).

**Experiments: (1) Future visits prediction and (2) data imputation.** For both experiments, models are trained with a 5-folds cross-validation. In the prediction experiment, we estimate, in the out-of-train fold, the individual parameters $z_i$ on the first $n_f$ visits of patients and predict on their very last visit. In the imputation experiment, in the out-of-train fold, we estimate the individual parameters $z_i$ on all the patients visits except $n_r$ visits drawn uniformly in its set of observations and predict on these $n_r$ removed visits. We evaluate the performance of estimations with the Mean Absolute Errors (MAE) on the 5 test-sets, depending on the value of $n_r$ and $n_f$ varying from 1 to 2.

**Results.** The table 1 demonstrates that GB outperforms DCM except on ABETA 3.2, no matter the value of $n_f$ and $n_r$. Looking at the figure 3 c, the MMSE trajectory learnt by GB has a more exponential shape compared to the DCM and the HIPP trajectory has been transformed from a nearly linear curve into a piece-wise linear curve (see Supp. material C for all figures), the inter-patient variability is changed correspondingly (see sub-figure a and b). It shows that there are different trajectories behaviors that DCM cannot take into account. With GB, at the beginning of the disease we observe that HIPP volume decreases while MMSE stays constant, which is clinically coherent. However, GB fails to generalize on ABETA because of the small number of data (576 missing values out of 909 visits). When few information is available logistic prior for trajectories' shape is already almost optimal. It has been remarked that GB makes better predictions than DCM especially on patients with high scores, this fact is worth mentioning if we want to later improve predictions with ensemble methods [35]. We observe that the algorithm begins to stabilize after 4 steps which motivates that $N_{\text{comp}} = 6$ is a good trade-off between under-fitting and over-fitting. Now, we should assess the proposed method on a more general cohort including MCI and cognitively normal patients.

**TADPOLE Challenge. [26]** The TADPOLE training set is composed of data from the first three ADNI phases (ADNI 1, ADNI GO and ADNI 2). It includes approximately 1500 features acquired from 1737 subjects (957 males and 780 females) during 12.741 visits for at most 22 distinct time points, between 2003 and 2017. Challengers were evaluated on their prospective predictions on enrolled individuals rolling-over for ADNI next phase, regarding the Alzheimer's Disease Assessment Scale Cognitive Score (ADAS-Cog13) and the volume of lateral ventricles normalized by the intracranial volume (VENTS), with the MAE metric. The final test set, disclosed in 2019 (time to prediction is about 2 years), is composed of 223 follow-up visits. This challenge allows for a fair comparison of the prediction performances presented here against 56 alternative methods. These methods include penalized regression, linear mixed-effect models, recurrent neural networks, and multi-task learning [26]. Since the challenge was closed, more algorithms have been presented for the same prediction task, using the same dataset, including [36] and [37] using deep RNNs, and [38] using random forests. The forecast of ventricle volumes (VENTS) and cognitive decline (ADAS-cog) presented in this paper have smaller mean absolute errors than all these competing methods.

We compared GB and DCM to the best Challenger. For the training, we select all patients having at least 2 visits and being amyloid-positive (ABETA < 977 pg/mL [31]), it left us 765 patients

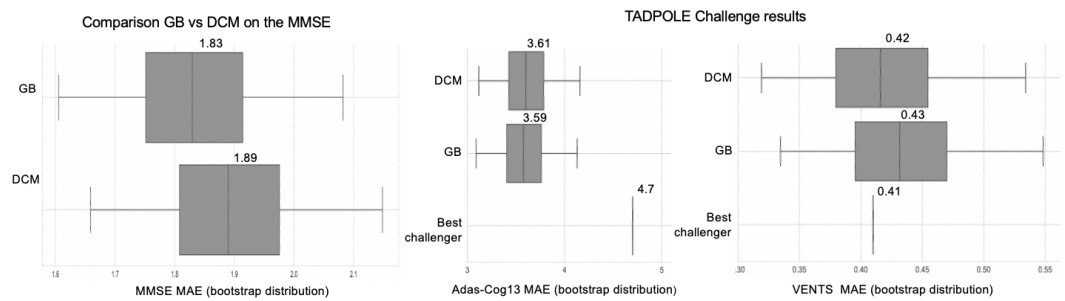

Figure 4: Boxplots after bootstrapping the prediction errors on the different scores. (For the boxplot : Whiskers=[2.5,97.5] %, Box=[25,50,75]%)

with 5.5 visits on average. We use the observations of ADAS-Cog13, MMSE, HIPP, VENTS and CSF P-TAU to calibrate the models with a 5-folds cross-validation. We apply the same transformation as previously to pre-process HIPP, VENTS and to CSF P-TAU (Box-Cox transform, quantile clipping and min-max rescaling) as well as MMSE and ADAS-Cog13 (only affine-rescaling since scores are already bounded). For GB, we take $N(\sigma = 0.24)$ , $N_{\text{comp}} = 4$ and $g_{\text{init}} = g^{\text{sig}}$ with 2 principals directions for space-shifts, but 3 for DCM.

**Results**    To know whether our results are statistically significant in the absence of other challengers' errors sample, we derive confidence intervals of DCM and GB's MAE by bootstrapping our error sample (10000 discounted draws). On figure 4, we observe that DCM and GB outperform the best challenger on ADAS-Cog13 and stay competitive on VENTS. This fact shows the potential of the geometric approach to offer interpretable and flexible models. Regarding the effect of learning the metric, GB seems to be better than DCM on ADAS-Cog13 and MMSE but not on VENTS. Comparing with the previous experience, it is likely that GB focuses more on the AD patients profile compared to MCI and controls. Indeed AD patients are representative of the disease progression whereas MCI and control subjects are seen by the model as slower disease progressors (delayed in time with a large time-shift $\tau_i$ and slow-paced with a low acceleration factor $\alpha_i$).

## 4   Conclusion

Riemannian metric learning applied to mixed-effect models enables to reach a sensible trade-off between flexibility and interpretability in disease progression modeling. The proposed approach allows us to disentangle time and space variability while learning the inter-patient variability and the average trajectory from the data. Without data scarcity, it proves to be efficient on homogeneous cohort for improving predictions on imaging and clinical biomarkers and suggests promising perspectives to handle heterogeneous cohorts with ensemble methods and mixture models. This work can be pursued by searching theoretical guarantees for the optimization process and practical guidelines for selecting hyper-parameters with empirical estimators. This model has the potential to better monitor patients and select them in clinical trials, although we need to pay particular attention to the representativity of minority groups in the training data set.

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
