# Supplementary document.

**Samuel Gruffaz**
Inria Paris
Ecole normale supérieure Paris-Saclay
samuel.gruffaz@ens-paris-saclay.fr

**Pierre-Emmanuel Poulet**
Paris Brain Institute
Inria Paris
pierre-emmanuel.poulet@inria.fr

**Etienne Maheux**
Paris Brain Institute
Inria Paris
etienne.maheux@icm-institute.org

**Bruno Jedynak**[*]
Maseeh Professor of Mathematical Sciences
Fariborz Maseeh Hall, room 464P
Portland, OR 97201
bruno.jedynak@pdx.edu

**Stanley Durrleman**[†]
Paris Brain Institute
Inria Paris
Inserm, CNRS, Sorbonne University
stanley.durrleman@inria.fr

for the Alzheimer's Disease
Neuroimaging Initiative [‡]

## A Key concepts of Riemannian geometry.

In this part, we don't claim to expose a rigorous nor comprehensive presentation of Riemannian geometry. Our goal is to clarify the notion of geodesics and parallel transport for the understanding of our model (see the book [1] for a comprehensive introduction). We assume that the reader has basic knowledge on manifolds and tangent spaces. Nevertheless, for sake of simplicity, we choose $\mathbb{M}$ to be an open subset of $\mathbb{R}^d$ instead of working with a general manifold.

**Riemannian manifold**

**Definition 1.** *If for all point $p$ in $\mathbb{M}$ $g_p(.,.)$ is a scalar product on $T\mathbb{M}_p$ and $p \to g_p$ is smooth, then $(\mathbb{M}, g)$ is a Riemannian manifold. $g$ is called the Riemannian metric.*

The notion of Riemannian manifold generalizes the usual notions of geometry to manifold. Remark that in an Euclidean space $\forall p \in \mathbb{R}^d$, $g_p = \langle .,.\rangle$, thus, a Riemannian metric encodes a richer geometry than an Euclidean geometry. An important notion in Riemannian geometry is the geodesic, which is the "shortest" path (in the sens of the Riemannian metric) between two points of the manifold.

---

[*]This work was partially supported by NIH R01AG027161 and R01EY032284.

[†]This research has also received funding from the program "Investissements d'avenir" ANR-10-IAIHU-06. This work was also funded in part by the French government under management of Agence Nationale de la Recherche as part of the "Investissements d'avenir" program, reference ANR-19-P3IA-0001 (PRAIRIE 3IA Institute) and by ANR under the joint programme in neurodegenerative diseases (JPND) ANR-19-JPW2-000 (E-DADS).

[‡]Data used in preparation of this article were obtained from the Alzheimer's Disease Neuroimaging Initiative (ADNI) database (adni.loni.usc.edu). As such, the investigators within the ADNI contributed to the design and implementation of ADNI and/or provided data but did not participate in analysis or writing of this report. A complete listing of ADNI investigators can be found at: http://adni.loni.usc.edu/wp-content/uploads/how_to_apply/ADNI_Acknowledgement_List.pdf

**Definition 2.** *Given a Riemannian manifold, we define the Riemannian distance $d$ such that*

$$\forall p_1, p_2 \in \mathbb{M}, \ d(p_1, p_2) = \inf_{\gamma:[0,1]\to\mathbb{M}, \gamma(0)=p_1, \gamma(1)=p_2} \int_0^1 \sqrt{g_{\gamma(t)}(\dot{\gamma}(t), \dot{\gamma}(t))} \, \mathrm{d}t \qquad (1)$$

*If $\gamma^*$ minimize the previous problem, $\gamma^*$ is called geodesic.*

In the Euclidean space, geodesics are straight lines such that $\ddot{\gamma} = 0$. We want to have such a characterization in Riemannian space. To get it, we need to know how to compute second order trajectory in a Riemannian space (to apply method of variational calculus in the previous minimization problem 1). Therefore, we should present the notion of connection.

**Connection** When we work with trajectories $\gamma : ]-1, 1[ \to \mathbb{M}$ with their values in a manifold $\mathbb{M}$, we can compute their first order derivative for each time $\dot{\gamma}(t) = \lim_{\epsilon\to 0} \frac{\gamma(t+\epsilon)-\gamma(t)}{\epsilon}$ and this element belong to the tangent space $T\mathbb{M}_{\gamma(t)}$ (by definition). The tangent spaces $(T\mathbb{M}_{\gamma(t)})$ are vector spaces with the same dimension (the codimension of the manifold), nevertheless, they could be all geometrically different (think of the tangent vector spaces associated to the sphere $\mathcal{S}^2$). This is why, it is not possible to define a "classic" second order derivative for $\gamma$, indeed, $\ddot{\gamma}(t) = \lim_{\epsilon\to 0} \frac{\dot{\gamma}(t+\epsilon)-\dot{\gamma}(t)}{\epsilon}$ can be undefined since $\dot{\gamma}(t+\epsilon)$ and $\dot{\gamma}(t)$ belong to two possibly different vector spaces (no guarantee on the addition stability). If $\mathbb{M}$ is an euclidean space $\mathbb{R}^d$, $(T\mathbb{M}_x)_{x\in\mathbb{R}^d}$ are all "geometrically" equal, that's why we can define derivative with order higher than one. In the general case, we need to *connect* the different tangent spaces to establish a correspondence between the value of a vector $v_1$ in $T\mathbb{M}_{p_1}$ and $v_2$ in $T\mathbb{M}_{p_2}$ to compare them.

To do it, we reduce the problem to the vectors of the basis of each tangent space $((e_i(p))$ the basis of $T\mathbb{M}_p$ that we define for all $p$ in $\mathbb{M}$) and we define how the basis is varying locally with an affine *connection* $\nabla$, in other words, how the tangents spaces are changing locally. Precisely, in each point $p$, the affine connection $\nabla$ takes two vector fields $v : p \in \mathbb{M} \to v(p) \in T\mathbb{M}_p$, $w : p \in \mathbb{M} \to w(p) \in T\mathbb{M}_p$ and returns a vector field $\nabla_v w$ (a vector field is a function taking a point $p \in \mathbb{M}$ and returning a vector in $T\mathbb{M}_p$) and follows three properties:

- left-linearity, for all three vector field $X_1, X_2$ and $Y$:
$$\forall f, g \in C^\infty(\mathbb{M}, \mathbb{R}), \forall p \in \mathbb{M}, \nabla_{fX_1+gX_2}Y(p) = f(p)\nabla_X Y(p) + g(p)\nabla_{X_2}Y(p)$$

- the right-Leibniz-linearity, for all two vector field $X, Y$ and $Y$:
$$\forall f \in C^\infty(\mathbb{M}, \mathbb{R}), \forall p \in \mathbb{M}, \nabla_X(fY)(p) = f(p)\nabla_{X_1}Y(p) + X(f)(p)Y(p)$$
where $X(f)(p) = \lim_{\epsilon\to 0} \frac{f(\gamma(\epsilon))-f(p)}{\epsilon}$ and $\gamma : ]-1, 1[ \to \mathbb{M}$ is a diffentiable curve such that $\gamma(0) = p, \dot{\gamma}(0) = X(p)$.

- the $\mathbb{R}-$right-linearity, forall three vector field $X, Y_1, Y_2$:
$$\forall a, b \in \mathbb{R}, \forall p \in \mathbb{M}, \nabla_X(aY_1 + bY_2)(p) = a\nabla_X Y_1(p) + b\nabla_X Y_2(p)$$

To define $\nabla_v w$, we set for all point $p$ in $\mathbb{M}$ $\nabla_{e_i}e_j(p) = \sum_k \Gamma_{i,j}^k(p)e_k(p)$ where the coefficients $\Gamma_{i,j}^k(p) \in \mathbb{R}$ are characterizing the connection thanks to the linearity properties, they are called the Christoffel's symbols. Thus, if we write $\forall p \in \mathbb{M}, w(p) = \sum_i \psi_i(p)e_i(p), v(p) = \sum_i \phi_i(p)e_i(p)$ with $\phi_i, \psi_i \in C^\infty(\mathbb{M}, \mathbb{R})$:

$$\forall p \in \mathbb{M}, \nabla_v(w)(p) = \sum_k (v(\psi_k)(p) + \sum_{i,j} \phi_i(p)\psi_j(p)\Gamma_{i,j}^k(p))e_k(p)$$

Now, we can compute second order derivative by considering that $\nabla_{\dot{\gamma}}\dot{\gamma}$ ($\dot{\gamma}$ seen as the restriction of a vector field) is the analogue of $\ddot{\gamma}$.

$$\nabla_{\dot{\gamma}}\dot{\gamma}(\gamma(t)) = \sum_k (\ddot{\gamma}_k(t) + \sum_{i,j} \Gamma_{i,j}^k(\gamma(t))\dot{\gamma}_i(t)\dot{\gamma}_j(t))e_k(\gamma(t))$$

where $\ddot{\gamma}_k(t) = \lim_{\epsilon\to 0} \frac{\dot{\gamma}_k(t+\epsilon)-\dot{\gamma}_k(t)}{\epsilon}$ if $\dot{\gamma}(t) = \sum_i \dot{\gamma}_i(t)e_i(\gamma(t))$. Applying Variational calculus method to the problem 1, we find as second order condition that $\nabla_{\dot{\gamma}}\dot{\gamma} = 0$ which gives:

$$\ddot{\gamma}_k(t) + \sum_{i,j} \Gamma_{i,j}^k(\gamma(t))\dot{\gamma}_i(t)\dot{\gamma}_j(t) = 0 \qquad (2)$$

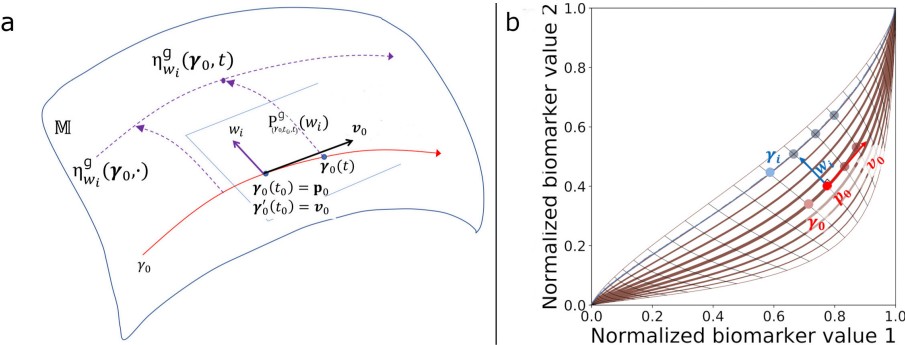

Figure 1: a) sketch of exp-paralellisation in an abstract manifold. b) exp-parallelisation for the unit square as a manifold with the metric $g_{(p_1,p_2)}((x_1,x_2),(y_1,y_2)) := \frac{x_1 y_1}{p_1^2(1-p_1)^2} + \frac{x_2 y_2}{p_2^2(1-p_2)^2}$. The average trajectory in red and an individual trajectory in blue or purple.

This second order Ordinary Differential Equations (ODE) has solutions as soon as the Christoffel symboles are smooth enough and the ODE can be characterized with the Cauchy conditions in 0 ($\gamma(0) = p$, $\dot\gamma(0) = v$). Thus, the geodesics are characterized and we can define the Riemannian exponential:

**Definition 3.** *The Riemannian exponential is* $\mathrm{Exp} : (p,v) \in \mathbb{M} \times T\mathbb{M} \to \mathrm{Exp}_p(v) \in \mathbb{M}$ *where* $\mathrm{Exp}_p(v)$ *is the point that is reached at time 1 by the geodesic starting at p with velocity v solution of the ODE 2.*

*We say that a Riemannian space is complete if the map :* $\mathrm{Exp}_p : T\mathbb{M} \to \mathbb{M}$ *is well-defined for all point p in the manifold* $\mathbb{M}$ *(the ODE admits solutions defined on the open interval* $\mathbb{R}$*).*

The Riemannian exponential enables to manipulate geodesics by looking at solutions of ODEs 2 which is easier than looking at solutions of a minimization problem 1. Remark that in the Euclidean case $\mathrm{Exp}_p(v) = p + v$.

Another notion which is essential in geometry is the notion of Parallel transport. It enables to define parallel vectors in tangents space along a trajectory according to a given connection.

**Definition 4.** *The parallel transport of a vector* $X_0 \in T_{\gamma(t_0)}\mathbb{M}$ *on a given curve* $\gamma$ *is a time indexed family of vectors* $X(t) \in T_{\gamma(t)}\mathbb{M}$ *which satisfies* $\nabla_{\dot\gamma(t)}X(t) = 0$ *and* $X(t_0) = X_0$. $\nabla_{\dot\gamma(t)}X(t) = 0$ *is an ODE on* $X = \sum_k X_k e_k$ *with the Cauchy condition* $X(t_0) = X_0$ *:*

$$\ddot X_k(t) + \sum_{i,j}\Gamma_{i,j}^k(\gamma(t))\dot\gamma_i(t)X_j(t) = 0 \tag{3}$$

*We denote* $P_{\gamma,t_0,t}(X_0)$ *the isometry mapping* $X_0$ *to* $X(t)$.

Remark that in the Euclidean space $X(t) = X_0$ (parallel vector are colinear).

Equipped with this notion of parallel vector, now we see how to generate trajectories parallel according to a transformation called the "exp-parallelisation".

Let $(\mathbb{M}, g)$ denotes as a geodisically complete Riemannian manifold equipped with its Levi-Cevita connection $\nabla^g$ (a connection induced by the metric $g$ with interesting properties).

**Definition 5.** *Let* $\gamma$ *be a curve on* $\mathbb{M}$ *defined for all time, a time-point* $t_0 \in \mathbb{R}$ *and a vector* $w \in T_{\gamma(t_0)}\mathbb{M}$, $w \neq 0$. *One defines the curve* $s \to \eta_w^g(\gamma, s)$, *the exp-parallelisation of the curve* $\gamma$, *as:*

$$\eta_w^g(\gamma, s) = \mathrm{Exp}_{\gamma(t_0)}^g(P_{(\gamma,t_0,s)}^g(w))$$

Intuitively, the curve $\eta_w^g(\gamma, s)$ is a shift from $\gamma$ in the direction $w$ at $t_0$. At $t_0$, the point constructed is simply the exponential of $w$ starting from $\gamma(t_0)$. For the rest of the curve, the only thing changing is that we don't take directly $w$ (as it leaves in $T\mathbb{M}_{\gamma(t_0)}$), but the result of the transport of this vector $w$ parallel to the curve $\gamma$.

## A.1 Proofs

**Proposition A.1** (Pushforward metrics). *Provided $(\mathbb{M}, g)$ a Riemannian space, $N$ a manifold and $\phi : \mathbb{M} \to N$ a $C^1$ diffeomorphism, we can equip $N$ with the Riemannian metric $g^\phi$ defined as:*

$$\forall\, p \in N, \;\; \forall w, v \in TN_p, \;\; g_p^\phi(w, v) = g_{\phi^{-1}(p)}(\mathrm{d}\,\phi^{-1}(p).w, \mathrm{d}\,\phi^{-1}(p).v)$$

*Moreover, $\phi$ is an isometry, which implies that for all $(m, v)$ in $T\mathbb{M}$ and $\gamma :]-1, 1[\to \mathbb{M}$ a differentiable curve:*

$$\mathrm{Exp}_{\phi(m)}^{g^\phi}(\mathrm{d}\,\phi(m).v) = \phi \circ \mathrm{Exp}_m^g(v), \;\; P_{\phi\circ\gamma,t_0,s}^{g^\phi}(\mathrm{d}\,\phi(\gamma(t_0)).v) = \mathrm{d}\,\phi(\gamma(s)).P_{\gamma,t_0,s}^g(v)$$

*Proof.* see [2] for the push-forward and see [3] at page 60 for the implication. $\square$

**Proposition A.2.** *if $\phi = \mathrm{id} + f$, with $f$ a bounded function in $C_(^1\mathbb{R}^d, \mathbb{R}^d)$ such that $|||\,\mathrm{d}\,f|||_\infty < 1$, then $\phi$ is a $C^1$ diffeomorphism.*

*Proof.* Let $f$ be a function verifying the previous conditions. We know that $\forall A \in \mathcal{M}_d(\mathbb{R})$ s.t $|||A||| < 1$, $I + A$ is invertible. Thus, we have $\forall x \in \mathbb{R}^d, \mathrm{d}\,\phi(x) \in Gl_d(\mathbb{R})$. Therefore, $\phi$, being $C^1$, is a local $C^1$ diffeomorphism.

Let's show that $\phi$ is injective. If $\phi(x) = \phi(y)$ with $x, y \in \mathbb{R}^d$ distinct, we have $|x - y| = |f(y) - f(x)|$ which is absurd by the theorem of finite increments since $|||\,\mathrm{d}\,f|||_\infty < 1$. Let's show that $\phi$ is surjective by showing that $\Im(\phi) = \{\phi(x), x \in \mathbb{R}^d\}$ is an open ensemble and closed in $\mathbb{R}^d$. $\Im(\phi)$ is open since $f$ is a local $C^1$ diffeomorphism. To show that $\Im(\phi)$ is closed, we consider a sequence $(y_n = f(x_n))$ of $\Im(\phi)$ converging to $y$ and we want to prove the existence of $x \in \mathbb{R}^d$ s.t $\phi(x) = y$. If $(x_n)$ is bounded, we can consider a converging subsequence $(x'_n)$ to a point $x$ and by continuity we have $\phi(x) = y$. If $(x_n)$ is not bounded, we have $|y_n - x_n| = |f(x_n)|$ which is absurd by hypothesis. Therefore, $\phi$ is bijective and a local $C^1$ diffeomorphism, $\phi$ is a global $C^1$ diffeomorphism.

$\square$

**Lemma A.1.** *If $k$ is a kernel such that $k(x, y) = g(x - y)\,\mathrm{Id}$ with $g \in C^2(\mathbb{R}^d, \mathbb{R})$ and $h(x, y) = g(x - y)$ is a bounded kernel, then $\forall f \in \mathcal{H}$ s.t $||f||_\mathcal{H} < \frac{1}{\sqrt{|\nabla g(0)|}}, \forall x \in M, \;\; |||\,\mathrm{d}\,f(x)||| < 1$.*

*Examples:*

- *The Gaussian kernel $g(x) := \exp(-\frac{|x|^2}{2\sigma^2})$, $\sigma > 0$ $\frac{1}{|\nabla g(0)|} = \frac{\sigma}{\sqrt{d}}$.*

- *The Sobolev kernel $g(x) := \frac{1}{(1+\frac{|x|}{2\sigma^2})^a}$, $\sigma > 0, a > d$, $\frac{1}{|\nabla g(0)|} = \frac{\sigma}{a\sqrt{d}}$*

*Proof.*

$$\partial_i f(x).\alpha = \langle \partial_i k(x, .)\alpha, f \rangle$$

$$\mathrm{d}\,f(x)^T.\alpha = (\langle \partial_i k(x, .)\alpha, f \rangle)_i$$

$$||\,\mathrm{d}\,f(x)^T.\alpha||_2^2 = \sum_{i=1}^d \langle \partial_i k(x, .)\alpha, f \rangle^2$$

$$\forall \alpha \in S^{d-1}, ||\,\mathrm{d}\,f(x)^T.\alpha||_2^2 \leq ||f||_\mathcal{H}^2 \sum_{i=1}^d ||\partial_i k(x, .)\alpha||_\mathcal{H}^2$$

$$\forall \alpha \in S^{d-1}, |||\,\mathrm{d}\,f(x)|||^2 \leq ||f||_\mathcal{H}^2 \sum_{i=1}^d ||\partial_i k(x, .)\alpha||_\mathcal{H}^2$$

We can enforce $||f||_\mathcal{H}^2 \sum_{i=1}^d ||\partial_i k(x, .)\alpha||_\mathcal{H}^2 < 1$ by assuming:

$$\forall \alpha \in S^{d-1}, ||f||_\mathcal{H}^2 < \frac{1}{\sum_{i=1}^d ||\partial_i k(x, .)\alpha||_\mathcal{H}^2}$$

Using the properties of a $C^2$ kernel, we have:

$$||\partial_i k(x,.)\alpha||_H^2 = \alpha^T \partial_{i,i}^2 k(x,.)\alpha$$

If $k(x,y) = h(x,y)I_d$, we have:

$$\alpha^T \partial_i^2 k(x,x)\alpha = ||\alpha||^2 \partial_{i,i}^2 h(x,x) = \partial_{i,i}^2 h(x,x)$$

Then

$$||f||_H^2 < \frac{1}{\Delta h(x,x)}$$

With $h(x,y) = g(x-y)$, we have

$$||f||_H^2 < \frac{1}{\Delta g(0)}$$

$\square$

# B   Complexity

We comment here on the time complexity of the Alternating Optimization algorithm for the Geodesics Bending model.

In each of the $N_{\text{comp}}$ steps of the algorithm, we perform $n_{\text{MCMC}}$ steps of the MCMC-SAEM algorithm and one optimization of the metric. The time complexity $T$ of the MCMC-SAEM for the mixed-effect model is already a function of the dimension of the observations, the number of principal directions of variance and the total number of observations. Our goal is not to evaluate the time complexity of the mixed-effect estimation but to understand the impact of metric learning on this time. The GB method adds a cost to the computation of the geodesics, which is proportional to the number of compositions for the diffeomorphism, thus to the number of the current iteration, and also proportional to the number of control points $N(\sigma)$. So for each MCMC-SAEM optimization in an outer step, we have a time complexity of $\mathcal{O}(n_{\text{MCMC}}N(\sigma)N_{\text{comp}}T)$. The time complexity $C$ of the metric optimization depends on the solver, but it depends on the kernel matrix in the quadratic problem which has a $N(\sigma) \times N(\sigma)$ size. However in our code the computation time of this step was negligible compared to the MCMC-SAEM time (about 1s vs several minutes). So the overall complexity of the algorithm is $\mathcal{O}(n_{\text{MCMC}}N(\sigma)N_{\text{comp}}^2T + N_{\text{comp}}C)$.

# C   Experiment details

## C.1   Convergence and Stability parameters.

For both experiments, we selected as fixed effect

$$t_0 = 70, v_0 = (0.2,0), \log(p_0) = (0,1), \sigma_\tau^2 = 5, \sigma_\xi^2 = 0.1$$

and for the principal direction $A_1$, we took the vector orthonormal to $v_0$.

We chose different values for the coordinates of velocities $v_0$ and initial position $p_0$ in order to see the effect of asymmetric trajectories. The results are similar no matter the choice of the fixed effect as soon as values are not "extreme".

The most important parameters are the number of subjects $N_{pat}$, the average number of visits $n_{\min}^t + n_\delta^t/2$, the noise intensity $\sigma_{\text{noise}}$ and the standard deviation for time generation $\sigma_t$, they influence directly the quality of the dataset. We try to select them to ressemble the TADPOLE dataset.

## C.2   Choice of $\sigma^2$

On Synthetic data, we observed that GB is likely to overfit if $\sigma < 3\sigma_{\text{noise}}$. Thus, we have looked for $\sigma$ in [0.2,0.5] to reduce the time of computations for experiments on real dataset since we estimate the noise to be around 0.05 on average. On the Alzheimer cohort, we have seen that $\sigma = 0.24$ is a better choice than $0.2, 0.3, 0.4$ but it is not statistically significant. For TADPOLE, we get the same conclusion. Nevertheless, we think that it would be interesting to select $\sigma$ dimension-wise as multidmensional data my not respect the isotropy of a standard Gaussian kernel. In that way, we can choose a smaller $\sigma$ for features with faster variations and conversely.

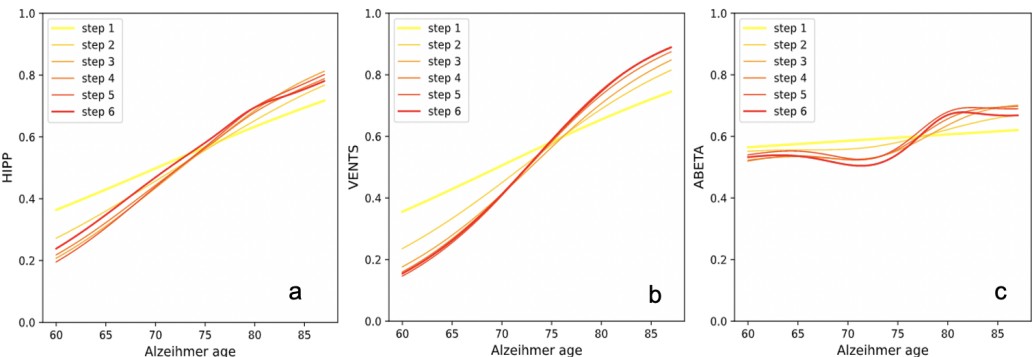

Figure 2: (a) The average trajectory $\gamma_0$ evolving through steps of GB is shown for HIPP, VENTS and ABETA in the Alzheimer cohort experiment.

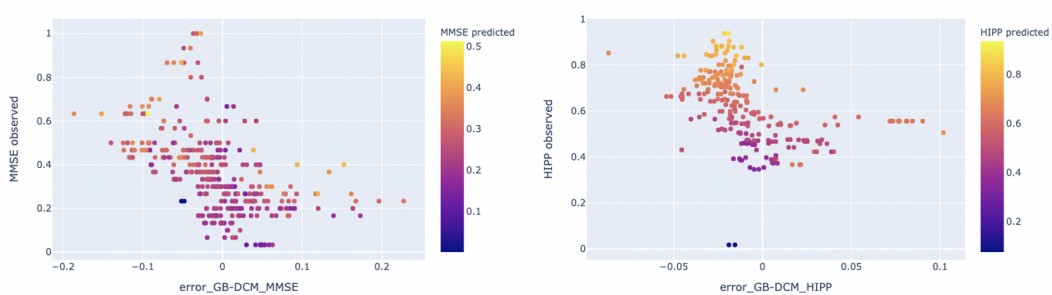

Figure 3: On the x-axis, we measure the difference of predictions errors between GB and DCM, if x is negative it means that GB is better than DCM and conversely. On the y-axis, we measure the "real" score values $y_{i,j}$ and we color points according to the value predicted by GB. If there is a correlation between the real and the predicted value, it motivates the use of ensemble methods to improve predictions. On the left figure, the values are associated to the MMSE predictions and on the right figure to the HIPP.

### C.3 Variation of the effects $\theta$ through steps of GB.

We have observed the variation of the fixed-effect from one step to another in Geodesics Bending. In general, the biggest variation is done between the first and the second step. When the iterations begin to stabilize, there is always a variation which is due to the exploration of MCMC-SAEM, but it does not affect a lot the global shape of the trajectory.

### C.4 Compositions' impact on neuro-psychological scores.

For the Alzheimer cohort experiment, we report the others figures where the effect of compositions is shown fig. 2. It is interesting to observe that the scores' trajectory are mainly changed on their velocity $v_0$ and little curved at their extremity. We can interpret it as follows : the transformation of asymptots at the extremity enables the velocity $v_0$ to increase the slope at the center. For example on ABETA, the asymptots in 1 is reduced and the asymptots in 0 is increased, it enables the middle part of the trajectory to have a higher slope to fit the "transition" between two states. This phenomenon seems to improve predictions for patients with high scores (meaning that GB is better on long term predictions), see figure 3.