# OpenReview forum: "Learning Riemannian metric for disease progression modeling"
_NeurIPS.cc/2021/Conference — NeurIPS 2021 Poster_

### Official Review · Reviewer_FMHL · 2021-07-06

**Rating:** 7
**Confidence:** 4

**Summary:**

The paper presents a generalization of the linear mixed effect model to a Riemannian manifold. Instead of using a heuristically designed Riemannian metric, the paper proposes to learn the metric directly from data. This is achieved by learning a push forward function that maps the flat space to curved space. The method is validated on the ADNI data in metrics from the TADPOLE challenge.

**Limitations And Societal Impact:**

Lack of connection to other disease progression modeling works.

**Main Review:**

The Riemannian formulation of the problem, although falling to a relatively narrow research field, seems to be an interesting extension of prior established theory. Using an RKHS function to parameterize the diffeomorphic push forward mapping is a standard reasonable construction. There are several points that I would like the authors to further clarify.

1. I wonder how such a Riemannian concept/setting is essential for modeling this kind of problem. How do authors view the relationship between Riemannian modeling vs. directly using more flexible parameterization in flat space to learn trajectories (is it feasible to direct use RKHS in the context of Euclidean Linear mixed effect model)? The proposed method relies on geodesics on the manifold, i.e. linear modeling on a non-linear space. What are the pros and cons of that practice compared to non-linear modeling in flat space?

2. In line with the above comment, the Riemannian setup makes it an ‘isolated' approach that can only be compared to one baseline GB. I encourage the authors to discuss their proposal in a broader context and really try to see whether is proposal approach compares favorably to many other branches of methods in progression modeling.

3. Can the authors confirm that training/testing subjects used herein are the same as the TADPOLE challenge. I'm confused by "To know whether our results are statistically significant in the absence of other challengers’ errors sample, we derive confidence intervals of..."  Is MMSE also provided as a training feature in the TADPOLE challenge? I remember TADPOLE predicts three outcomes, but only two are given in this paper (ADAS, VENTS).

minor: I don't understand why an isometric $\phi$ can induce curvature preservation.

**Time Spent Reviewing:**

3 hours

---

> ### Author Response · Authors · 2021-08-10
> **Review answer**
>
> We thank you for your insightful feedback on our work.
>
>    1) We agree with the reviewer that there are several ways to turn a linear model into a non-linear one. We believe that transforming a flat observation space (or latent space) into a Riemannian manifold offers specific advantages. First, this construction is generic and applies to any data that can be represented in a manifold. This property allows the DCM technique (and hence our GB method) to be applied not only for biomarker data but also for structured data: images (https://www.frontiersin.org/articles/10.3389/fneur.2018.00235/full) and shapes (https://link.springer.com/article/10.1007/s11263-020-01343-w), thus avoiding the need to derive a new model for each data type. Second, the Riemannian metric offers a way to control the non-linearity in a theoretically well-grounded framework. The Riemannian structure offers specific tools, such as the possibility to define a spatiotemporal coordinate system in the observation space to position the progression of different subjects. In particular, it allows learning dynamical parameters (age at onset, acceleration factors) together with subject-specific progression parameters while ensuring the model to be identifiable (see https://jmlr.csail.mit.edu/papers/volume18/17-197/17-197.pdf for a detailed discussion). Eventually, a geodesic on a Riemannian manifold is a particular case of a Hamiltonian flow, where the Hamiltonian is given by the Riemannian metric. We propose here to learn the dynamics of the biomarker changes by learning the metric and hence the Hamiltonian of a dynamical system. Our work, therefore, relates to the Hamiltonian and normalizing flows that become popular in the machine learning community (see e.g.  https://arxiv.org/abs/1909.13789 or https://arxiv.org/abs/1908.09257.) The counterpart of this more abstract construction is an overhead cost in understanding the method which might not be justified when applied on the simpler cases, such as the progression of a single one-dimensional biomarker.
>
>    2) Our method predicts the whole trajectory of change of the biomarkers. It contrasts therefore with black-box machine learning methods that predict the value of the biomarkers after a fixed time interval (say after 1 year), a problem that can be formulated as a regression problem. Recurrent neural networks have been proposed to predict a sequence of biomarker values. Such methods, however, need to cope with missing values, a limited and variable number of observations per patient, a variable time interval between observations, and a very variable pattern of change depending on disease stage and subject. Mixed-effect models make several assumptions about the pattern of progression. In particular, they assume that the progression falls within a parametric family of curves of small dimension, e.g. a linear curve or a logistic curve with unknown parameters.
>
>    3) In the AD experiment (the first one in our paper) the patients are indeed the same as for the TADPOLE challenge, but the model used in the TADPOLE challenge was a new one (fit only on the TADPOLE train set) so there is no data leakage. About the confusion : "To know whether our results are statistically significant in the absence of other challengers’ errors sample, we derive confidence intervals of..." means that we only have access to mean absolute errors at the group level and not the errors for each observation. Therefore, we could derive a confidence interval of the MAE for DCM and GB only. We will clarify these points in the final version of the paper.
>     MMSE is provided as a training feature among many others, but submissions were assessed only in the prediction of ADAS and VENTS (see https://tadpole.grand-challenge.org/Results/).

---

### Official Review · Reviewer_kAFz · 2021-07-15

**Rating:** 7
**Confidence:** 3

**Summary:**

This work proposes a Riemannian metric learning applied to mixed-effect models which enables a balance between flexibility and interpretability in disease progression modeling. The proposed approach allows us to disentangle time and space variability while learning the inter-patient variability and the average trajectory from the data. Main innovation is the way of avoiding the selection of Riemannian metric through empirical arguments by proposing a semi-parametric method using a Reproducing Kernel Hilbert Space (RKHS).

**Ethical Concerns:**

No concerns. ADNI is a verified open database.

**Limitations And Societal Impact:**


1)Please put brief limitations and explanations why (in your opinion) what you propose works better than something probably more established.
Typically I kindly suggest to add paragraph before Section 4 which explains some technical aspects about the reason why you find these Results (page 9). For example, what makes your approach to outperform on ADAS - Cog 13 and not on VENTS. Is it by chance?Is it the dataset and the distributions behind?
Please mention also the methods with whom you compare.

2)
There is a nice bibliography that guides well the reader. However, I did not see any mention in the Multitasking learning approaches for disease progression prediction which have been highly popular nowadays and fairly less "complex" (in my opinion) for the multidisciplinary world. Indicatively you can initially check the works  of:

Zhou et al 2011 A Multi-Task Learning Formulation for Predicting

Zhou et al 2012 Modeling disease progression via fused sparse group lasso,”

Cao et al 2017 “Sparse shared structure based multi-task learning for MRI based cognitive performance prediction of Alzheimer’s disease
 or
Suk e al 2016 Deep sparse multi-task learning for feature selection in Alzheimer’s disease diagnosis

It would be nice to (at least) discuss a little bit the pros and cons.

Probably some of these approaches are already included in the TADPOLE challenge. It is good to know moe details about who wins where and the level of winning (extreme outperformance or minor?) etc.


No Negative Societal Impact. I would rather say that there is only positive impact with his work.

**Main Review:**

Originality:
The work is part of a larger effort to predict the disease progress. It offers  a promising and a flexible method and especially the Geodesic Bending (authors' name) is a clear originality of this work.

Quality: The quality is good. There is a nice balance between theoretical grounding and experiments. Supplementary material helped a lot.
Clarity: The paper is pretty clear. I was struggling to understand that you will also use DCM in the results which creates confusion.  Please clarify in the introductory section (it is there but it is not clear) what you will compare with what.

Sgnificance: It is a nice paper that improves the way that the Riemannian  metric is chosen with an elegant way.

**Time Spent Reviewing:**

2

---

> ### Author Response · Authors · 2021-08-10
> **Review answer**
>
> We appreciate your efforts in assessing our paper and thank you for your feedback.
>
> We will edit the introduction to better clarify what will be done in the paper in terms of comparison, as you and several other reviewers were concerned about it.
> Regarding your two points:
>
> ### First point
> Our method starts with a prior shape of the progression curve, for instance, a logistic or a linear function as in the DCM approach. It then bends the curve to better fit data. This bending occurs only in the age range where there are enough observations (usual for Bayesian methods). Therefore, the forecast of biomarkers far in the future might not work well if the asymptotic shape of the curve greatly varies among subjects. This is probably more the case for the volume of the ventricles in contrast to the ADAS-Cog assessment.
>
> One of the main limitations of our approach comes from the fact that we need to learn the metric in addition to the parameters of the DCM model. The dimension is controlled by the number of control points, the kernel parameter, and the number of compositions. We agree that more work is needed to further evaluate the method in addition to the encouraging results that we present here on both synthetic and real data.
>
> Our method predicts the whole trajectory of change of the biomarkers. It contrasts therefore with black-box machine learning methods that predict the value of the biomarkers after a fixed time interval (say after 1 year), a problem that can be formulated as a regression problem.
> Recurrent neural networks have been proposed to predict a sequence of biomarker values. Such methods, however, need to cope with missing values, a limited and variable number of observations per patient, a variable time interval between observations, and a very variable pattern of change depending on disease stage and subject.
>
> Mixed-effect models make several assumptions about the pattern of progression. In particular, they assume that the progression falls within a parametric family of curves of small dimension, e.g. a linear curve or a logistic curve with unknown parameters. Our contribution is exactly to extend this approach by broadening the class of functions.
>     We will elaborate more on the explanation of the strengths and limitations of our approach in the final version of the paper.
>
>
> ### Second point
> We thank the reviewer for pointing out these references, which we will add to the final version of the paper. As a matter of fact, submissions to the TADPOLE challenge included multi-task learning methods, so we did compare our results with such techniques. We will better present the different types of methods that were included in the TADPOLE challenge in the final version of the paper.

---

### Official Review · Reviewer_EBmu · 2021-07-16

**Rating:** 6
**Confidence:** 2

**Summary:**

Linear mixed-effect models provide a natural baseline for estimating disease progression using longitudinal data. This paper applies Riemannian metric learning to mixed-effect models to improve their flexibility while keeping the interpretability. Specifically, the metric is learned as the push-forward of the Euclidean metric by a diffeomorphism that is estimated iteratively as the composition of radial basis functions belonging to a reproducible kernel Hilbert space. The metric update allows to improve the forecasting of imaging and clinical biomarkers in the Alzheimer's Disease Neuroimaging Initiative (ADNI) cohort. Experimental results on synthetic data and on a real dataset shows that it performs better than or comparably to the 56 methods benchmarked in the TADPOLE challenge on the task of predicting patient's biomarker progression.

**Ethical Concerns:**

None.

**Limitations And Societal Impact:**

Yes.

**Main Review:**

Comments:
1. Figure 1 needs to be explained in more detail.

2. Language issues and others
Line 96: There is a redundant article "the".
Line 103: "depicted on 1" should be "depicted in (1)" (using LaTex command $\eqref{}$).
Line 103: $]0,1[$ should be written as $[0,1]$.
Line 120: "not necessary true" should be "not necessarily true".
Line 123: "in 1" should be "in (1)".
Eq. (2): The middle term is missing the Euclidean norm symbol $| |$.
Line 132: The word "time" is repeated twice.
Line 133: $]-1,1[$ should be written as $[-1,1]$.
Line 148: $]0,1[$ should be written as $[0,1]$.
Line 149: "a RKHS" should be "an RKHS".
Line 153: What is the symbol "Id"?
Line 157: Should $|x|$ be $|x|^2$? Please cite the source of the Sobolev kernel. I cannot find the description of the Sobolev kernel in the references e.g., [20].
Line 160: "problem 2" should be "problem (2)".
Line 162: "the error of reconstruction 2" should be "the error of reconstruction in (2)".
Line 162: $w_{i,j}^*$ should be $w_{i,j}$ (i.e., no asterisk).
Line 196: "on a Git in a near future" should be "on Github in the near future".
Line 212: "both experiment(s)".
Line 218: "on 2" should be "in Figure 2".
Table 1 caption: "record" should be "recorded".

**Time Spent Reviewing:**

4

---

> ### Author Response · Authors · 2021-08-10
> **Review answer**
>
> We would like to answer both of your points :
>
> 1) Regarding Figure 1, it represents how the exp-parallelization works. It is similar to the Euclidean parallelization where each translation is replaced by the exponential mapping of the manifold to adapt to the structure of the data. The figure shows how to construct subject-specific trajectories as a parallel curve to a central geodesic. This construction defines a spatiotemporal coordinate system on the manifold to position the progression of each subject. We will make sure to add a more detailed caption.
>
> 2) We will take particular attention to the typos and correct the mathematical notations in the final version of the paper. Regarding the Sobolev kernel, it is also known as the generalized T-student kernel (p.24 "kernels methods : Generalisations, scalability ans towards the future of machine learning , Oxford University"). We will clarify this point.

---

### Official Review · Reviewer_W3mj · 2021-07-18

**Rating:** 6
**Confidence:** 4

**Summary:**

This paper learns Riemannian metric from data for disease progression. The proposed method has been validated on two datasets. This paper also demonstrates several theoretical results of the proposed methods.

**Limitations And Societal Impact:**

In term of novelty and performance improvement, this paper meets the publication requirement.

**Main Review:**

This paper proposed a novel method for learning metric from data. The paper provided some proofs of the proposed methods. But, this paper should give more details of the experimental setting, the explanation of the Figures, which will make the paper more clarity. The experimental results on two dataset shows the results for disease progression prediction. Some comments and suggestions are listed as follows:
1.In line 97: There are some typo.
2.In Figure 2: The authors should give the more explanation of the figure.
3.In Line 224 – 225: How to choose the σ. I guess the author select the best value
for the proposed method by using the cross-validation.  But, I think it is a bad choice. The dataset in your experiments on the ADNI dataset just separate the ADNI dataset into training and validation sets. The choice in the validation sets may produce bad results on the real test data.
4.In Line 237-245: Author should give more rigorous experiments to demonstrate the generalization of the proposed methods.
5.Line 260-361: There are some same subjects in the ADNI1, ADNI GO, ADNI2, and ADNI3 datasets.  Is there a data leakage?  Are you only using AD data?
6.There are many methods proposed for metric learning and disease progression prediction. Please compared with the state-of-the-art methods.
7.The claimed contribution of this paper is the interpretability. But the paper lacks the descriptions of the model interpretability and result interpretability.
8.Please give the time and space complexity analysis of the proposed method.


**Time Spent Reviewing:**

3

---

> ### Author Response · Authors · 2021-08-10
> **Review answer**
>
> We would like to address each of your points :
>    1) We will carefully edit the paper to correct several typos
>    2) Figure 3 is about the description of the benefits of learning the metric compared to DCM. Fort instance, we observe that through compositions of the deformation (steps 1,2,3..) in figure c, the learned average trajectory reveals some pattern in the data that DCM (step 0) was unable to catch. For the figure a and b, we reach the same conclusion when comparing the differents exp-parallelization with the metric of DCM and GB. We will add such explanations in the final version of the paper.
>    3) The value of the kernel parameter $\sigma$ has been chosen after having tried several values in the synthetic experiments. It was not adjusted by cross-validation on the real data set. This section on Synthetic experiments is presented for illustrating the potential of our method and for giving guidelines for selecting the hyper-parameter sigma to avoid underfitting/over-fitting. We remark that we should take $\sigma$ in [0.1,0.5] and that $\sigma\approx 0.22$ is a good compromise when the parameters of data generation empirically match actual datasets of interest. A more advanced hyperparameter search would require further investigations.
> For the experiments with real datasets, we did not perform cross-validation to avoid further time complexity. Instead, we used the guidelines of the Synthetic experiments by trying $\sigma=0.11,0.22,0.5$ and evaluating the results on a validation set.
>    4) We understand that the paragraph on lines 237-245 is misleading. Synthetic experiments were made for a qualitative evaluation of the method such as the impact of parameters on results. Qualitative evaluation is done using ADNI data, in particular in the framework of the TADPOLE challenge. Learning the metric is an operation which can be seen as a layer atop the DCM, thus the generalization comes from the flexibility of the Riemannian framework, allowing to work on various types of data as was done with DCM (cf "Spatiotemporal Propagation of the Cortical Atrophy: Population and Individual Patterns").
>    5) There was no data leakage as patients have the same identifier in all data sets (ADNI1, ADNI-GO, ADNI2, and ADNI 3). The split between train and test sets was made based on these patients' IDs after pooling data from all data sets. In our experiment (AD experiment), we selected only patients declared AD since their first visit in order to have the most homogeneous cohort.
>    6) For clarification purposes, note that the comparison with other disease progression methods as well as a variety of alternatives is done thanks to the TADPOLE challenge. Indeed, this challenge allows for a fair comparison of the prediction performances presented here against 56 alternative methods. These methods include penalized regression, linear mixed-effect models, recurrent neural networks, and multi-task learning. Ref. 24 in the manuscript is the main reference for analyzing the results. Since the challenge was closed, more algorithms have been presented for the same prediction task, using the same dataset, including  \url{https://www.sciencedirect.com/science/article/pii/S1053811920306893} and \url{https://link.springer.com/chapter/10.1007/978-3-030-32251-9_19} using deep RNNs, and   \url{https://journals.plos.org/plosone/article?id=10.1371/journal.pone.0211558} using random forests. The forecast of ventricle volumes (VENTS) and cognitive decline (ADAS-cog) presented in this paper have smaller mean absolute errors than all these competing methods. We will better clarify the scope of this comparison in the final version of the paper.
>    7) Our method is interpretable in the sense that it predicts the progression trajectory of the biomarkers. It contrasts therefore with other methods that predict the value of the biomarker or a diagnostic category at a given time-point, say in 1 year in the future. Interpretability is inherited from the DCM method, on top of which we develop the geodesic bending (GB) method. In this paper, we focus more on our main contribution which is the learning of the shape of the progression trajectory in a semi-parametric way.
>    8) Time complexity is mentioned in the last paragraph of the method section. Space complexity: we store a rectangular matrix with dimensions the number of observations times the number of control points. It is far from being an issue in our applications. We will add this information about space complexity in the final version of the paper.

---

### Author Response · Authors · 2021-08-10
**Common answer to reviews**

Dear reviewers,

We are thankful for your work and kind comments on our manuscript following our submission for publication. We have highlighted several common criticisms from your reviews and gathered them in this answer. More specific answers will be directly available for each review.

### Quantitative comparison with state-of-the-art

For clarification purposes, note that the comparison with other disease progression methods as well as a variety of alternatives is done thanks to the TADPOLE challenge. Indeed, this challenge allows for a fair comparison of the prediction performances presented here against 56 alternative methods. These methods include penalized regression, linear mixed-effect models, recurrent neural networks, and multi-task learning. Ref. 24 in the manuscript is the main reference for analyzing the results. Since the challenge was closed, more algorithms have been presented for the same prediction task, using the same dataset, including  https://www.sciencedirect.com/science/article/pii/S1053811920306893 and https://link.springer.com/chapter/10.1007/978-3-030-32251-9_19 using deep RNNs, and   https://journals.plos.org/plosone/article?id=10.1371/journal.pone.0211558 using random forests.

The forecast of ventricle volumes (VENTS) and cognitive decline (ADAS-cog) presented in this paper have smaller mean absolute errors than all these competing methods.

We will better clarify the scope of this comparison in the final version of the paper.

### Qualitative comparison with state-of-the-art

Our method predicts the whole trajectory of change of the biomarkers. It contrasts therefore with black-box type machine learning methods that predict the value of the biomarkers after a fixed time interval (say after 1 year), a problem that can be formulated as a regression problem.

Recurrent neural networks have been proposed to predict a sequence of biomarker values. Such methods, however, need to cope with missing values, a limited and variable number of observations per patient, a variable time interval between observations, and a very variable pattern of change depending on disease stage and subject.

Mixed-effect models make several assumptions about the pattern of progression. In particular, they assume that the progression falls within a parametric family of curves of small dimension, e.g. a linear curve or a logistic curve with unknown parameters. This is the case for instance of the DCM approach where the linear or logistic curve are seen as the solution of an Hamiltonian flow (a geodesic is a particular case of an Hamiltonian flow) with a given Hamiltonian function. Our main contribution is to learn the metric of the Riemannian manifold, and therefore the Hamiltonian, to better learn the dynamics of progression of the biomarker. In this respect, our contribution relates to the techniques of Hamiltonian and normalizing flows that are becoming popular in the machine learning community, see e.g. https://arxiv.org/abs/1909.13789 or https://arxiv.org/abs/1908.09257.

We will clarify the positioning of our approach with respect to the state of the art along these lines, and add a dedicated "related work" section in the introduction.

---

### Decision · Program_Chairs · 2021-09-28

**Decision:**

Accept (Poster)

**Comment:**

This work proposes a Riemannian metric learning applied to mixed-effect models which enables a balance between flexibility and interpretability in disease progression modeling. This work is original in predicting the disease progress. The theoretical grounding and experiments are well provided. Overall, the paper is well written. There are some suggestions, such as related works and details of experimental setting, as well as some typos in notations. Please carefully consider all suggestions from reviewers in the final version.

**Consistency Experiment:**

NeurIPS has a long history of experimentation. In 2014, NeurIPS ran an experiment in which 10% of submissions were reviewed by two independent committees to quantify the randomness in the review process. This year, we repeated a variant of this experiment to see how the quality of the review process has changed over time.  This paper was part of the experiment and was therefore assigned to two committees (consisting of reviewers, an Area Chair, and a Senior Area Chair) that reached independent decisions.  If both committees made the same recommendation, this recommendation was followed. If a single committee recommended acceptance, the paper was accepted (with the exception of a few cases in which the other committee identified what we considered a fatal flaw, e.g., an error in a key result).

This copy’s committee reached the following decision: **Accept (Poster)**

The other committee assigned to the paper recommended **Reject**.  You can find the other set of reviews, along with any follow up discussion with the authors here:
https://openreview.net/forum?id=qZpOqPbwhy